# Clinical and Immunological Impact of Ocrelizumab Extended Interval Dosing in Multiple Sclerosis: A Single-Center, Real-World Experience

**DOI:** 10.3390/ijms25105353

**Published:** 2024-05-14

**Authors:** Martina Nasello, Valeria Zancan, Virginia Rinaldi, Antonio Marrone, Roberta Reniè, Selene Diamant, Martina Marconi, Lorenzo Le Mura, Marco Salvetti, Maria Chiara Buscarinu, Gianmarco Bellucci

**Affiliations:** 1Department of Neuroscience, Mental Health and Sensory Organs (NESMOS), Sapienza University of Rome, 00185 Rome, Italy; martina.nasello@uniroma1.it (M.N.); valeria.zancan@uniroma1.it (V.Z.); mchiara.buscarinu@gmail.com (M.C.B.);; 2IRCCS Istituto Neurologico Mediterraneo Neuromed, 86077 Pozzilli, Italy

**Keywords:** multiple sclerosis, B cells, disease modifying treatments, ocrelizumab, extended dosing, immunoglobulins

## Abstract

Ocrelizumab (OCR), an anti-CD20 monoclonal antibody, is approved for treating relapsing remitting (RR) and primary progressive (PP) multiple sclerosis (MS). The standard interval dosing (SID) regimen requires intravenous infusions every six months. Experience of extended dosing due to COVID-19 pandemic-related issues suggests that this strategy may provide comparable efficacy while reducing treatment burden and healthcare costs. This study aimed to evaluate clinical effectiveness, changes in B- and T-cell count, and immunoglobulin dynamics associated with extended interval dosing (EID) of ocrelizumab in a real-world setting. We retrospectively included RRMS or PPMS patients treated with OCR that had already received two OCR cycles and with at least 6 months of follow up after the last infusion. EID was defined as a ≥4 weeks delay compared to SID. Clinical outcomes were occurrence of relapses, MRI activity, 6-months confirmed disability progression (CDP) and their combination (No Evidence of Disease Activity, NEDA-3). We also evaluated changes in CD19+ B cell count, CD4+ and CD8+ T cell count, immunoglobulin titers, and occurrence of hypogammaglobulinemia (hypo-Ig). Frequency tests, multivariate regression models, and survival analysis were applied as appropriate. We analyzed data on 93 subjects (75.3% RRMS) for a total of 389 infusions (272 SID, 117 EID). Clinical and MRI activity, CDP, and NEDA 3 did not significantly differ between EID and SID. EID was associated with lower rates of B-cell depletion. T-cell dynamics and incidence of hypo-Ig were comparable following EID and SID. Hypo-IgG at index infusion was associated with further occurrence of hypo-IgG; male sex and hypo-IgM at index infusion were independently associated with hypo-IgM. In conclusion, OCR EID does not impact MS clinical and radiological outcomes, although it interferes with B-cell dynamics. These findings provide support for a tailored schedule of OCR in MS.

## 1. Introduction

Ocrelizumab (OCR) is a recombinant humanized monoclonal antibody targeting CD20-expressing cells, approved for the treatment of both relapsing remitting (RR) [1] and primary progressive (PP) multiple sclerosis (MS) [2]. The therapeutic effect of OCR is related to an antibody-dependent and complement-dependent depletion of CD20+ B cells (pre-B cells, immature B cells, and memory B cells) [3]. B cells are pivotal in the development and progression of MS, and B-cell depleting treatments tackle several B-cell driven pathogenetic mechanisms: antigen presentation to T cells; release of pro-inflammatory cytokines like IL-6, IL-17, and TNF; and establishment of ectopic lymphoid follicles in the meninges [3]. Recent studies have shown how the depletion of a subset of CD20+ T cells may also contribute to OCR therapeutic functioning [4,5].

The standardized therapeutic regimen of OCR consists of an induction phase (of two infusions of 300 mg in 14 days) and a maintenance phase of 600 mg every 6 months. Recent evidence has shown that B-cell depletion starts 2 weeks after infusion and can last for more than 6 months [6]; likewise, clinical effects span over the dosing schedule, supporting the positioning of OCR within the immune-reconstitution therapies [7].

Despite the high clinical efficacy of this therapy, it can be associated with adverse effects, mostly infections related to secondary lymphopenia and hypogammaglobulinemia (hypo-Ig) [8,9].

To optimize the risk–benefit ratio, several studies during the recent SARS-CoV-2 pandemics have focused on the possibility of extending the interval between anti-CD20 infusions [10,11,12,13,14], also personalizing the timing based on peripheral B-cell count and depletion status [15,16,17].

However, none of these works performed a complete analysis of the effects of OCR Extended Interval Dosing (EID) on B cells, T cells, and immunoglobulins (Ig). Here, we evaluated such effects together with clinical and radiological changes in a real-world setting.

## 2. Results

### 2.1. Study Cohort and Baseline Characteristics

Figure 1 shows study design (see Section 4). We screened 130 patients with MS (pwMS) receiving OCR at our center. We excluded 37 subjects due to insufficient data or short follow up (median age 42 years; 48.6% male; 81% RRMS; median EDSS 3.0; mean ARR 0.68; median disease duration 4.49 years; DMT-naïve 27%; 45.9% had clinical and 59.5% radiological activity in the previous year). Thus, we retrospectively collected and analyzed data from 93 pwMS treated with OCR.

Table 1 shows the demographic and clinical characteristics at baseline. A total of 64 patients received at least 1 EID, with baseline characteristics not significantly different from the SID group, except for a longer mean follow-up time (*p* < 0.001). Fourteen patients received 2 EID, 10 patients received 3 EID, 5 patients received 4 EID, and 1 patient received cumulatively 5 EID. The median dosing interval in the EID group was 227 days (mean, 260 days). We monitored a total of 389 infusions and related 6-month follow ups, of which 272 were SID and 117 EID regimens. We tested differences in baseline features among MS phenotypes. RRMS showed higher baseline ARR (0.55 [0.72] vs. 0.08 [0.34], *p* < 0.001), as might be expected, while PPMS had higher EDSS at the time of starting OCR (5 [2] vs. 2.5 [3], *p* < 0.001). RRMS had higher rate of MRI activity (75.7% vs. 13%, *p* < 0.0001) and of clinical activity (62.9% vs. 17.4%, *p* < 0.001) in the year before starting OCR. Sex did not affect baseline characteristics. Expectedly, naïve patients had shorter disease duration (0.5 vs. 8.7 years, *p* < 0.0001) compared with pwMS switching from other DMTs.

We found a moderate, negative correlation between ARR and disease duration (rho = −0.54, *p* < 0.0001) that hold true also when controlling for age (rho = −0.50, *p* < 0.0001). Conversely, there was a positive correlation between baseline EDSS and disease duration (rho = 0.31, *p* = 0.002; rho = 0.23, *p* = 0.02 controlling for age) and inversely related to ARR (rho = −0.37, *p* = 0.0002) (Appendix A).

Table 2 shows the baseline values of lymphocytes and immunoglobulins. There were no significant differences between SID and EID groups. Total lymphocyte count was negatively correlated to the number of DMTs previously received (rho= −0.20, *p* = 0.04). IgG, IgM and IgA levels were positively interrelated (*p* < 0.0001) but not linked to B- and T-cell counts. Similarly, CD8+, CD4+, and CD19+ levels were positively correlated. Sex and MS phenotype did not influence baseline lab values. We did not find any difference in lymphocyte subsets and Ig values among naïve patients and those switching from various DMTs, except for a trend between higher CD19+ levels in people switching from natalizumab (*p* = 0.06) (Appendix A).

### 2.2. Ocrelizumab EID Does Not Impact Clinical Outcomes

In the whole cohort, we observed a total of three clinical relapses, with no significant differences in the occurrence following SID or EID regimens (2 [0.7] %vs 1 [0.9%], *p* = 0.99). Disability progression occurrence did not differ significantly between the two regimens: 17 events (8.3%) following SID and 10 (10.9%) following EID (*p* = 0.51). MRI progression occurred in 21 subjects (13 [17.2%] EID, 8 [27.5%] SID), of which two pwMS experienced two distinct occurrences of MRI activity, again with no significant differences in the cumulative occurrence the two groups (*p* = 0.39) nor between the incidence following SID or EID (13 [4.7%] vs. 8 [6.8%], *p* = 0.46).

We performed regression analysis to identify factors independently associated with clinical activity, MRI activity, and loss of NEDA-3 (Table 3 and Table 4). Results showed the presence of a clinical relapse before the index infusion as an independent predictor of MRI activity (OR 3.37, CI 1.3–8.73, *p* = 0.01), while male sex showed protective effects (OR: 0.33, CI 0.14–0.90, *p* = 0.03). No significant predictor was identified for clinical activity or loss of NEDA-3. Neither B-cell depletion at index infusion nor EID were associated with clinical and radiological outcomes.

### 2.3. B-Cell Dynamics Are Affected by Extended Dosing without Clinical and Radiological Implications

Next, we evaluated CD19+ B-cell dynamics. As expected, there was a rapid drop in CD19+ count following the first OCR cycle. We observed a positive correlation between dosing interval duration and absolute CD19+ count, also controlling for B-cell count at previous infusion and cumulative OCR cycles (partial correlation, Spearman’s rho:0.23, *p* < 0.001, Figure 2A). Coherently, there was a lower rate of B-cell depletion following EID compared to SID (76.1% vs. 91.2%, *p* = 0.0008), with a higher CD19+ absolute count, independently from the OCR cycle number (Figure 2B,C). Consistently with the above regression analysis, we did not find significant differences in clinical and radiological events in B-depleted and non-depleted subjects (Appendix A).

In the whole cohort, we evaluated the B depletion status in relation to dosing interval through a single-arm survival analysis (Figure 2D,E). At 240 days, 74.3% (95%CI, 64.9–85.0) of subjects were B-depleted, dropping to 53.4% (40.1–71.0) at 300 days (i.e., a 4-month delay with respect to SID).

### 2.4. T Lymphocytes and Immunoglobulin Dynamics Are Not Significantly Different in OCR EID

During OCR therapy, we observed a slight but progressive decline in total lymphocyte count (rho= −0.15, *p* < 0.001) depending on the decrease in CD8+ circulating cells, while CD4+ lymphocytes were not significantly affected (Figure 3A–C). There was a significant correlation between CD8+ and CD4+ levels (rho = 0.27, *p* < 0.001). No correlation was observed between total lymphocytes, CD8+, CD4+, and dosing interval (*p* = 0.66, *p* = 0.38, *p* = 0.40 respectively) or CD19+ levels (*p* = 0.08, *p* = 0.11, *p* = 0.09 respectively). Indeed, there was no significant difference between EID and SID (Figure 3D–F).

Immunoglobulin levels progressively decreased during OCR therapy, with a steeper decline observed for IgM (Figure 4A–C). We found a positive inter-correlation between IgM and IgG values, IgA and IgG values, IgG titers, and CD19+ cell count (Figure 4D–F), while no correlation was seen with interval duration (Figure 4G–I). While IgM and IgG levels were not significantly different following EID or SID, IgA were slightly higher following SID (*p* = 0.04, Figure 4L–N). We observed a hypo-IgG cumulative incidence of 20.5% with an exposure-adjusted incidence rate (EAIR) of 0.92 per 100 patient years, a 25.8% cumulative incidence of hypo-IgM (EAIR = 1.17 per 100 patient-years), and 2.15% of hypo-IgA (EIAR 0.25). No difference was observed in the occurrence of hypo-Ig following EID and SID (Appendix A). We applied regression models to identify factors associated with hypo-Ig (Table 5 and Table 6). At multivariate analysis, hypo-IgG at index infusion was associated with further occurrence of hypo-IgG (OR 4.00, CI 1.40–12.01, *p* = 0.01). Male sex (OR 3.39, CI 1.65–6.96, *p* = 0.0009) and hypo-IgM at index infusion (OR 4.67, CI 2.11–10.67, *p* < 0.001) were independently associated with hypo-IgM.

## 3. Discussion

Here, we performed a retrospective analysis on pwMS receiving OCR at our center and evaluated clinical, radiological, and immunological outcomes following SID and EID. Although resulting in higher rates of CD19+ B-cell repopulation, ocrelizumab EID did not affect clinical and radiological outcomes. Adding to previous evidence [18], it supports the hypothesis that OCR long-lasting efficacy may go beyond B-memory depletion, through mechanisms that are preserved—if not even potentiated [19]—by EID or personalized dosing. T-cell and immunoglobulin dynamics were also unaffected.

Consistent with ours, most of the available retrospective studies highlighted no difference in clinical and radiological disease activity between SID and EID [13]. The note of caution came from Zanghì and colleagues, who identified an increased risk of MRI activity following EID [12]. Still, there was no impact on clinical activity, disability progression, and NEDA-3 status. Also, MRI activity was more frequent in people with a short disease duration, i.e., in the early phase of disease history, when the inflammatory activity tends to be more prominent. This caveat should be considered in planning EID. Despite the rarity of clinical and radiological events in our OCR-treated cohort, we found that the occurrence of a recent clinical relapse is an independent predictor of MRI progression, plausibly supporting the notion of “partial response” to OCR. Male sex showed protective effects towards MRI activity (OR 0.37), consistent with previous data showing fewer radiological signs of inflammation in male RRMS-SPMS [20].

In our cohort, EID was associated with a lower rate of B-cell depletion. This result is in line with previous data from Guerrieri and colleagues, who used the same absolute B depletion threshold (10 CD19+ cells/ μL) [10], and from Kumar and colleagues [17], who adopted a relative cutoff of 1%. Other studies [11], instead, did not detect significant differences between the groups. However, in all these works, CD19+ depletion was not linked to clinical and radiological outcomes. These results suggest that B-cell count may be insufficient to capture response to OCR and should be integrated to other biomarkers of immunological function or nervous tissue damage. It is reassuring that, in the prospective studies by Schuchmann and Zoe [13,15], tailored OCR extended dosing was not associated with higher NfL, but correlation with peripheral CD19+ was not assessed. Another option could be the use of other B-cell biomarkers, such as CD20, that captures B cells in a narrower window of maturation (from pre-B cells to B-memory subset) and CD27 that labels from B-memory cells to plasma cells [21]. Assessment of both CD19+ and CD20+ cells during OCR therapy showed similar dynamics and no clinical implications [10]. Personalized dosing of Rituximab based on the CD19+/CD27+ B-memory count was found to be efficient in disease activity reduction in an uncontrolled prospective study [16]. Additional studies with comprehensive immunophenotyping and more sensible clinical and radiological measures are awaited.

We identified a positive correlation between CD19+ and OCR dosing interval, expected based on known lymphocyte repopulation dynamics and consistent with previous reports [10,14,17]. We were also able to infer the probability of B-cell depletion status according to dosing interval (Figure 2D,E): based on our data, at 8 months post-infusion (240 days), the probability of having less than 10 CD19+ cells/uL is 74.3% (CI, 85–64.9%). This is consistent with observations by Mahmoud AbdelRazek and colleagues, who performed a monthly assessment of CD19+ in pwMS receiving OCR or RTX [22]. Even if they consider different cutoffs for depletion, both relative (<5%, <2%) and absolute (20 CD19/uL) cutoffs, the observed depletion rate in their cohort at 8 months (84.6% to 65.4%) is similar to our inference; differences emerge considering depletion rate at 10 months (their observed 33.0%, vs. our predicted 53.4%). Important to be considered is the more limited number of measurements at longer follow ups (N = 3 at month 11) in their study, compared with ours (N = 15 at month 11). Indeed, in repopulation-based dosing approaches, median dosing time ranged from 8.5 to 11.5 months [13,15]. Also, it is plausible that B repopulation kinetics may be also slower in patients with long-time treatment history.

We observed a modest effect of EID on CD8+ T cells (Figure 3B), independently of dosing interval. This effect might be a consequence of the depletion in the subpopulation of CD8+/CD20+ T cells, which has previously been linked to OCR clinical efficacy [23,24]. In a longitudinal study by Landi and collaborators [25], with a 12-month follow up, the T-cell depletion was related to a carry-over phenomenon of previous DMTs, especially fingolimod. We failed to identify such signal, at baseline and longitudinally (regression models). This could depend on the smaller sample size (e.g., 11 pwMS switching from fingolimod, compared with 52) or on a more specific OCR-dependent effect that may prevail at a longer follow-up time.

People with MS receiving anti-CD20 are at higher risk of developing hypo-Ig. In our cohort, the exposure-adjusted incidence of hypo-Ig was relatively low and in line with previous reports adopting similar definitions [26]. We did not detect a relationship between dosing interval and Ig levels, as has been described in other studies [27,28]. However, attempts are being made to define hypo-Ig’s risk factors in large cohorts to improve risk management [26].

The main strength of this study, compared with similar real-world evidence, is the completeness of the biological profiling of patients, with the availability of relevant data on lymphocytes’ subsets and Ig at different longitudinal timepoints during OCR treatment. The main limitations of the study are its retrospective observational design, in which blood testing was not performed at a fixed time; the relatively small number of subjects; the different follow-up duration between EID ans SID groups (that however could not influence the infusion-based statistical analysis); and the partially lacking information with respect to EID motivation and infectious events.

In conclusion, OCR EID is suitable in MS patients and does not impact clinical and radiological disease control. Prospective trials comparing EID to SID with integrated outcomes (such as unconventional MRI measures and fluid biomarkers of progression) are awaited to support our results.

## 4. Materials and Methods

### 4.1. Inclusion Criteria and Definitions

We included RRMS and PPMS patients receiving OCR that had (1) completed the first two administration cycles (2 × 300 mg, 2-weeks apart, and the second cycle of 600 mg) and (2) that had at least 6 months of follow up after the infusion (Figure 1).

EID was defined as a delay in OCR administration of at least 4 weeks, in line with other studies [10,11,12,17].

We reviewed and collected clinical, laboratory, and MRI data performed per clinical practice. Laboratory values are referred to the last test performed before the infusion.

Clinical activity was identified by the presence of relapses (new or exacerbating symptoms persisting for at least 24 h in the absence of fever or concurrent illness, at least 30 days after a previous relapse).

MRI activity was defined as the presence of T1-weighted gadolinium-enhancing lesions and/or new or enlarging T2-weighted lesions in a brain and spinal cord MRI.

Confirmed disability progression (CDP) was identified as a 6-month confirmed increase in the expanded disability status scale (EDSS) of 1.5 points if baseline EDSS was 0, 1.0 point if baseline EDSS was 1.0–5.5, and 0.5 points if baseline EDSS was >5.5.

The status of No Evidence of Disease Activity (NEDA) was qualified by the absence of all clinical activity, MRI activity, and CDP.

We also collected data on total lymphocyte count, CD8+, CD4+, CD19+ cells, and immunoglobulins. B-cell depletion was considered to be a CD19+ count <10 cells/mL.

Thresholds for hypo-Ig were considered as follows, in accordance with previous literature [29]: 700 mg/dl for IgG, 40 mg/dl for IgM, and 70 mg/dl for IgA.

### 4.2. Data Analysis

Statistical analyses were performed in Prism (version 9.5.1) and R (version 4.2.3). Variable distribution was assessed with the Shapiro–Wilk test. Continuous variables were compared with the Mann–Whitney U-test and Dunn’s corrected Kruskal–Wallis test. Categorical variables were compared with Fisher’s exact test. Spearman’s rho was calculated to assess correlations between continuous variables, and partial correlations were performed to control for potential confounding factors.

Univariate regression models were applied to identify predictors for clinical, radiological, and laboratory parameters (not applicable to hypo-IgA due to the low cumulative number of events). Covariates with *p* < 0.25 were included in multivariate models. A single-arm survival analysis was implemented to assess B-cell depletion status in relation with dosing interval (survminer package in R).

Outcomes were analyzed with an infusion-based approach (comparing outcomes following a regular or delayed infusion regimen; Figure 1).

## 5. Conclusions

In conclusion, this study provides real-world evidence that Ocrelizumab EID does not negatively affect MS clinical and radiological outcomes. As OCR-extended or biologically tailored dosing may be favorable in terms of safety, pharmacoeconomics, and in programming therapeutic de-escalation, more efforts in this direction are needed to ensure regulatory approval.

## Figures and Tables

**Figure 1 ijms-25-05353-f001:**
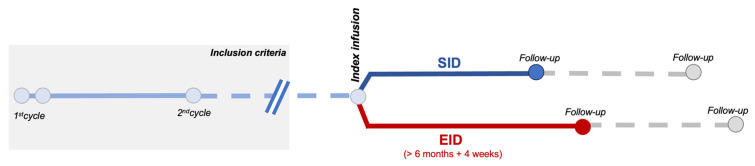
Schematic representation of the observational study design.

**Figure 2 ijms-25-05353-f002:**
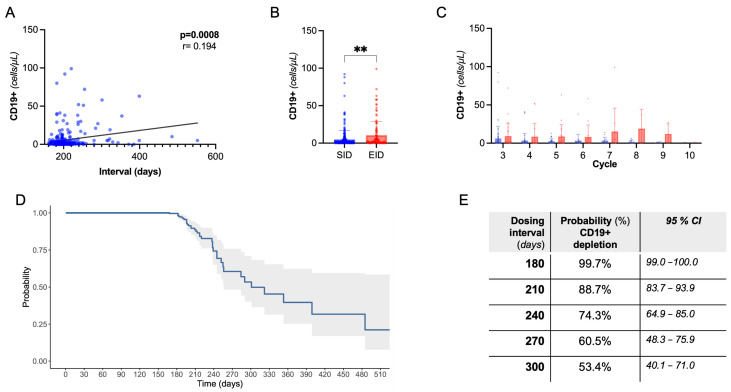
B-cell dynamics. (**A**) Spearman’s correlation of CD19+ cell count with dosing interval. (**B**) Peripheral CD19+ absolute count following EID vs. SID (Mann–Whitney U test); ** = significant with *p* < 0.01. (**C**) Peripheral CD19+ absolute count following EID vs. SID grouped by OCR cycle number. (**D**) Kaplan Meier curve showing the probability of B-cell depletion by dosing interval. (**E**) Referring to (**D**), probability and confidence interval by dosing time.

**Figure 3 ijms-25-05353-f003:**
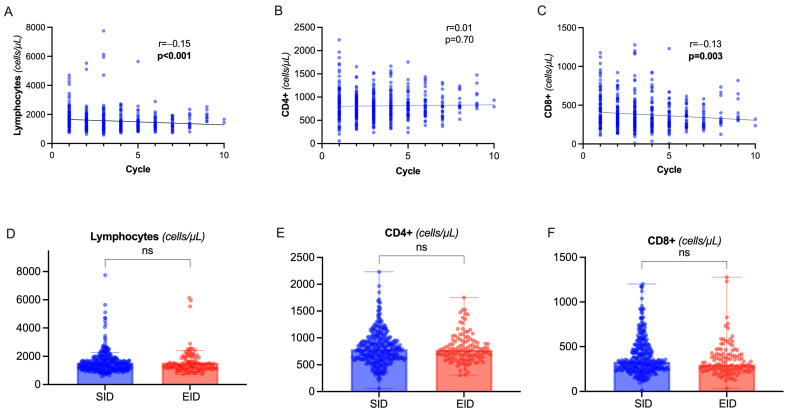
Total lymphocytes and T-cell dynamics. (**A**–**C**) Spearman’s correlation of total lymphocyte count (**A**), CD4+ (**B**), CD8+ (**C**) count with cumulative number of OCR cycles. (**D**–**F**) Differences in total lymphocyte count, CD4+, and CD8+ values between SID and EID (Mann–Whitney U test). Abbreviation: ns= non significant (*p* > 0.05).

**Figure 4 ijms-25-05353-f004:**
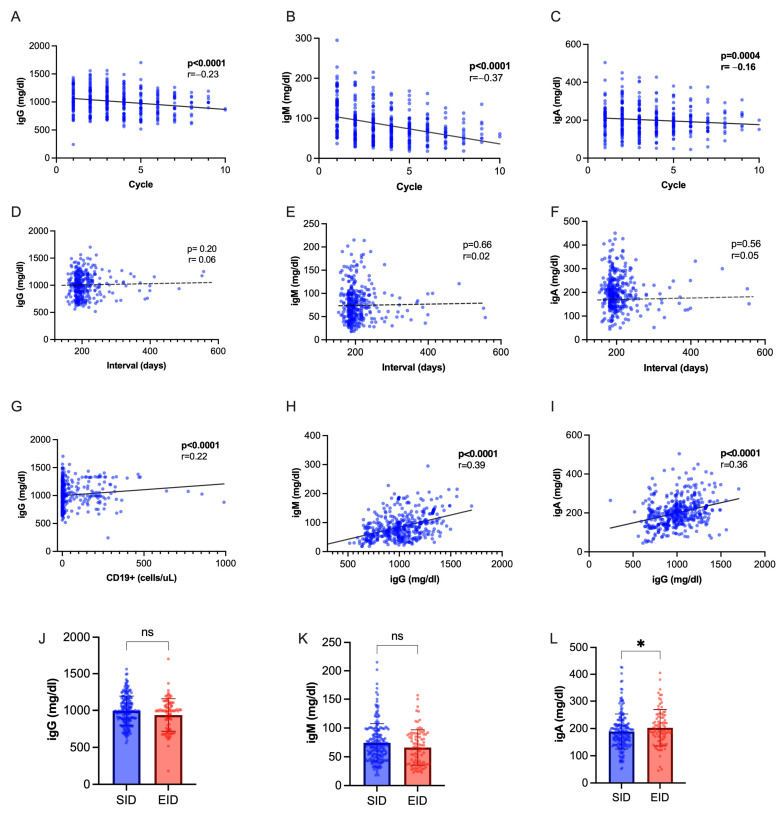
Immunoglobulin dynamics. (**A**–**C**) Spearman’s correlation of Ig titers with cumulative OCR cycles. (**D**–**F**) Spearman’s correlation of Ig titers with dosing interval. (**G**) Spearman’s correlation of IgG with CD19+ cell count. (**H**) Spearman’s correlation of IgG with IgM values. (**I**) Spearman’s correlation of IgG with IgA values. (**J**–**L**) IgG, IgM and IgA titers in SID vs. EID (Mann–Whitney U test). Abbreviations: * = *p* < 0.05; ns = not significant.

**Table 1 ijms-25-05353-t001:** Baseline demographic and clinical characteristics of the study population. Median (IQR) if not otherwise indicated; Mann–Whitney U test or Fisher exact test as appropriate. ARR: annualized relapse rate. Abbreviations are as follows. DMT: disease-modifying treatment. EDSS: expanded disability status scale. OCR: ocrelizumab. RRMS: relapsing-remitting MS. PPMS: primary-progressive MS.

	All (N = 93)	SID (N = 29)	EID (N = 64)	*p* Value
Age (years, mean ± SD)	46 ± 14	46 ± 14	45.5 ± 15.3	0.845
Male sex (N, %)	45 (48.4%)	13 (44.8%)	32 (50%)	0.66
RRMS (N, %) PPMS (N%)	70 (75.3%) 23 (24.7%)	22 (75.9%) 7 (24.1%)	48 (75%) 16 (25%)	1.0
Baseline EDSS	3.5 (3.5)	3.5 (3.0)	3.5 (3.5)	0.845
Baseline ARR	0.479 (0.764)	0.75 (1.04)	0.472 (0.58)	0.159
Disease duration (years, mean ± SD)	7.227 ± 13.3	2.97 ± 11.9	7.59 ± 13.7	0.06
Clinical activity in the year before starting OCR (N, %)	48 (51.6%)	14 (48.3%)	34 (53.1%)	0.54
MRI activity in the year before starting OCR (N,%)	53 (56.9%)	16 (55.2%)	40 (62.5%)	0.648
Cumulative EID	Median = 1	-	Median = 1	--
Naive (N, %)	23 (24.7%)	10 (34.5%)	13 (20.3%)	0.194
Last DMT				-
ALEMTUZUMAB	7	2	5	
AZATIOPRINE	1	0	1	
CLADRIBINE	2	0	2	
DMF	12	4	8	
FINGOLIMOD	11	1	10	
GA	10	5	5	
IFN	9	2	7	
NATALIZUMAB	11	5	6	
RITUXIMAB	2	0	2	
TERIFLUNOMIDE	5	0	5	
Reason to switch:				0.368
Efficacy (N, %)	58 (62.4%)	16 (55.1%)	42 (65.6%)	
Safety (N, %)	12 (12.9%)	3 (10.4%)	9 (14.1%)	
N Previous DMT	1 (2)	1 (2)	2 (2)	0.09
Follow-up duration after OCR initiation (months, mean ± SD)	27.2 ± 21.2	18.9 ± 6.7	33.4 ± 16.2	<0.001

**Table 2 ijms-25-05353-t002:** Baseline laboratory values. Mean (SD).

	All (N = 93)	SID (N = 29)	EID (N = 64)	*p* Value
Lymphocyte count (cells/µL)	1828 (779)	2131 (1007)	1690 (611)	0.069
CD4+ (cells/µL)	861 (384)	947 (433)	821 (357)	0.274
CD8+ (cells/µL)	446 (227)	495 (247)	424 (216)	0.174
CD19+ (cells/µL)	228 (160)	291 (225)	200 (110)	0.128
igG (mg/dL)	1065 (219)	1059 (229)	1068 (216)	0.917
igA (mg/dL)	211 (66)	212 (79)	211 (61)	0.993
igM (mg/dL)	121 (48)	117 (42)	123 (50)	0.502

**Table 3 ijms-25-05353-t003:** Regression analysis for clinical activity and MRI activity as outcomes.

	Clinical Activity				MRI Activity			
	Univariate Model		Multivariate Model		Univariate Model		Multivariate Model	
Covariate	OR (95% CI)	*p* Value	OR (95% CI)	*p* Value	OR (95% CI)	*p* Value	OR (95% CI)	*p* Value
Age	1.02 (0.91–1.17)	0.63			0.96 (0.92–1.01)	0.87		
Sex (male vs. female)	0.52 (0.02–5.43)	0.59			0.33 (0.07–1.14)	0.10	0.37 (0.14–0.90)	0.03
MS phenotype (PPMS vs. RRMS)	0.37 (0.01–79.0)	0.99			0.96 (0.25–3.12)	0.95		
No. previous DMTs	1.17 (0.52–2.06)	0.63			0.98 (0.62–1.40)	0.91		
Naïve at baseline (yes vs. no)	0.06 (0.02–13.2)	0.99			1.61 (0.42–5.29)	0.45		
EDSS at index infusion	1.08 (0.60–2.08)	0.80			1.01 (0.76–1.37)	0.92		
Cumulative OCR cycles before index infusion	0.49 (0.09–1.16)	0.23	0.02 (0.01–37)	0.99	0.86 (0.56–1.21)	0.43		
Disease Duration	1.17 (1.02–1.40)	0.04	1.10 (0.93–1.31)	0.27	1.01 (0.94–1.08)	0.74		
EID (yes vs. no)	1.16 (0.05–12.26)	0.90			1.11 (0.29–3.62)	0.87		
Cumulative EID before index infusion	0.77 (0.05–2.70)	0.77			1.44 (0.77–2.44)	0.21	1.08 (0.62–1.88)	0.77
Consecutive EID (yes vs. no)	1.64 (0.08–17.36)	0.69			2.03 (0.44–7.12)	0.32		
B-cell depletion at index infusion (no vs. yes)	1.26 (0.04–26.7)	0.99			1.45 (0.46–3.75)	0.47		
MRI activity before index infusion (yes vs. no)	20.0 (0.76–237.0)	0.04	6.56 (0.29–155)	0.23	1.75 (0.09–10.0)	0.61		
Clinical activity before index infusion (yes vs. no)	3.36 (0.15–35.70)	0.33			3.29 (1.28–7.82)	0.04	3.37 (1.30–8.73)	0.01

**Table 4 ijms-25-05353-t004:** Regression analysis for NEDA3.

	Loss of NEDA-3			
	Univariate Model		Multivariate Model	
Covariate	OR (95% CI)	*p* Value	OR (95% CI)	*p* Value
Age	0.97 (0.94–0.99)	0.04	0.98 (0.95–1.03)	0.57
Sex (male vs. female)	0.62 (0.34–1.11)	0.11	0.84 (0.42–1.72)	0.64
MS phenotype (PPMS vs. RRMS)	0.59 (0.29–1.12)	0.12	0.80 (0.33–1.91)	0.61
No. previous DMTs	1.05 (0.87–1.25)	0.56		
Naïve at baseline (yes vs. no)	0.94 (0.46–1.79)	0.85		
EDSS at index infusion	0.98 (0.85–1.13)	0.79		
Cumulative OCR cycles before index infusion	0.85 (0.71–1.01)	0.06	0.82 (0.67–0.99)	0.24
Disease Duration	1.01 (0.98–1.05)	0.48		
EID (yes vs. no)	1.35 (0.73–2.43)	0.32		
Cumulative EID before index infusion	0.93 (0.62–1.33)	0.70		
Consecutive EID (yes vs. no)	1.24 (0.63–2.31)	0.52		
B-cell depletion at index infusion (no vs. yes)	0.99 (0.41–2.12)	0.98		
MRI activity at previous cycle (yes vs. no)	1.01 (0.15–3.87)	0.98		
Clinical activity at previous cycle (yes vs. no)	3.35 (0.15–35.89)	0.33		

**Table 5 ijms-25-05353-t005:** Regression analysis for hypo-IgG. Abbreviations are as follows. DMT: disease-modifying treatment. EID: extended interval dosing. EDSS: expanded disability status scale. OCR: ocrelizumab. RRMS: relapsing-remitting MS. PPMS: primary-progressive MS. OR: odds ratio. CI: confidence interval.

	Occurrence of Hypo-igG			
	Univariate Model		Multivariate Model	
Covariate	OR (95% CI)	*p* Value	OR (95% CI)	*p* Value
Age	1.02 (0.98–1.06)	0.25		
Sex (male vs. female)	0.90 (0.42–1.91)	0.77		
MS phenotype (PPMS vs. RRMS)	0.55 (0.21–1.28)	0.19	0.46 (0.17–1.21)	0.11
No. previous DMTs	1.17 (0.92–1.47)	0.17	0.94 (0.71–1.25)	0.68
Naïve at baseline (yes vs. no)	0.77 (0.28–1.85)	0.58		
EDSS at index infusion	1.12 (0.92–1.37)	0.26		
Cumulative OCR cycles before index infusion	1.07 (0.81–1.31)	0.51		
Disease Duration	1.05 (1.01–1.10)	0.02	1.04 (0.98–1.09)	0.11
EID (yes vs. no)	1.03 (0.94–1.14)	0.53		
Cumulative EID before index infusion	0.99 (0.52–1.90)	0.90		
Consecutive EID (yes vs. no)	1.10 (0.63–1.71)	0.70		
B-cell depletion at index infusion (no vs. yes)	0.98 (0.28–2.70)	0.96		
Hypo-IgG at index infusion (yes vs. no)	4.44 (1.58–11.50)	0.002	4.00 (1.37–11.7)	0.01
Hypo-IgM at index infusion (yes vs. no)	1.01 (0.29–2.79)	0.98		
Hypo-IgA at previous infusion (yes vs. no)	3.01 (0.15–24.40)	0.35		

**Table 6 ijms-25-05353-t006:** Regression analysis for hypo-IgM. Abbreviations are as follows. DMT: disease-modifying treatment. EID: extended interval dosing. EDSS: expanded disability status scale. OCR: ocrelizumab. RRMS: relapsing-remitting MS. PPMS: primary-progressive MS. OR: odds ratio. CI: confidence interval.

	Occurrence of Hypo-igM			
	Univariate Model		Multivariate Model	
Covariate	OR (95% CI)	*p* Value	OR (95% CI)	*p* Value
Age	1.01 (0.98–1.04)	0.49		
Sex (male vs. female)	3.09 (1.61–6.21)	0.001	3.39 (1.65–6.96)	0.001
MS phenotype (PPMS vs. RRMS)	1.02 (0.53–1.93)	0.94		
No. previous DMTs	1.03 (0.83–1.25)	0.81		
Naïve at baseline (yes vs. no)	0.31 (0.10–0.75)	0.02	0.41 (0.15–1.16)	0.09
EDSS at index infusion	1.01 (0.87–1.17))	0.98		
Cumulative OCR cycles before index infusion	1.01 (0.84–1.20)	0.89		
Disease Duration	1.01 (0.97–1.05)	0.62		
EID (yes vs. no)	0.95 (0.51–1.78)	0.89		
Cumulative EID before index infusion	0.99 (0.26–2.38)	0.98		
Consecutive EID (yes vs. no)	1.03 (0.92–1.14	0.54		
B-cell depletion at index infusion (no vs. yes)	1.61 (0.96–1.51)	0.19	1.23 (0.54–2.80)	0.61
Hypo-IgG at index infusion (yes vs. no)	3.47 (1.42–8.45)	0.006	1.96 (0.71–5.41)	0.19
Hypo-IgM at index infusion (yes vs. no)	5.40 (2.60–11.26)	<0.001	4.67 (2.11–10.67)	<0.001
Hypo-IgA at index infusion (yes vs. no)	5.20 (0.71–37.77)	0.10	1.80 (0.18–11.70)	0.60

## Data Availability

The original contributions presented in the study are included in the article/Appendix A, further inquiries can be directed to the corresponding authors.

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
