# Peer review of "Clinical and Immunological Impact of Ocrelizumab Extended Interval Dosing in Multiple Sclerosis: A Single-Center, Real-World Experience"

_ijms, 2024, doi:10.3390/ijms25105353_

Round 1
Reviewer 1 Report
Comments and Suggestions for Authors
The manuscript entitled "Clinical and Immunological impact of Ocrelizumab Extended Interval Dosing in Multiple Sclerosis: A Single-Center, Real-World Experience" is an interesting original article that compares the efficacy and safety of standard and extended interval dosing of ocrelizumab. It is a retrospective observational study that provides also data on lymphocytes subsets and immunoglobulins count during therapy. It is an important issue for doctors treating patients with multiple sclerosis. The article is well written and scientifically sound. There are no severe concerns. There are only a few issues that could be addressed:
- some abbreviations are not explained, e.g. NEDA-3 in the Abstract (line 26)
- RRMS is characterized by the occurrence of relapses and PPMS - by a constant progression of disability, it is obvious that relapses occur in RRMS, not in PPMS (line 79)
- there a lot of typing mistakes, e.g. IgaA (line 148), in Table 2, in Fig. 3.
I think the article may be published after minor revision.
Comments on the Quality of English LanguageThere a lot of typing mistakes, e.g. IgaA (line 148), in Table 2, in Fig. 3, but otherwise English language provides no concerns.
Author Response
We thank the reviewers for their suggestions, which allowed us to improve the manuscript.
We addressed all the points. Please find below a point-by-point description of the revision.
REVIEWER 1
- some abbreviations are not explained, e.g. NEDA-3 in the Abstract (line 26)
R: We checked and explained abbreviations along the text.
- RRMS is characterized by the occurrence of relapses and PPMS - by a constant progression of disability, it is obvious that relapses occur in RRMS, not in PPMS (line 79)
R: We admit such observation in line 101.
- there a lot of typing mistakes, e.g. IgaA (line 148), in Table 2, in Fig. 3.
R: We corrected the typos along the text and in figures.

Reviewer 2 Report
Comments and Suggestions for Authors
Martina Nasello et al. Provides real-world evidence that occlizumab (OCR) with extended dosing interval (EID) does not affect multiple sclerosis (MS) clinical and radiological outcomes, which may provide new biological insights and future treatments for MS Potential strategies for sclerosis.
I believe readers in this area will benefit greatly from it, but the paper needs improvement to accept publication. Some minor revisions are listed below:
1. Authors should briefly describe a summary of the experimental methods used in this study.
2. Properties and OCR functionality should be described in more detail in the introduction.
3. Font sizes in figures should be consistent, and there should be font sizes that are too small to read clearly, as shown in Figure 3 and Supplementary Figure 1. Please make the necessary changes.
4. Limitations of the study are not without being addressed in detail. A description of the limitations would help improve the structural integrity of the article, please improve this section.
5. In the discussion, the authors did not indicate future research plans or directions. Please add it to complete the article.
Author Response
REVIEWER 2
- Authors should briefly describe a summary of the experimental methods used in this study.
- We experimental settings in detail in the Methods section and figure 1 summarizes the study design.
Properties and OCR functionality should be described in more detail in the introduction.
R: We implemented anti-CD20 mechanisms of action in lines 47-50.
Font sizes in figures should be consistent, and there should be font sizes that are too small to read clearly, as shown in Figure 3 and Supplementary Figure 1. Please make the necessary changes.
R: We homogenized font size in all the main figures and reformatted supplementary figure 1.
Limitations of the study are not without being addressed in detail. A description of the limitations would help improve the structural integrity of the article, please improve this section.
R: We improved a paragraph on study limitations in lines 334-338.
- In the discussion, the authors did not indicate future research plans or directions. Please add it to complete the article.
R: We mentioned future directions in lines 340-342.
